# Oral Health Status and Multiple Sclerosis: Classic and Non-Classic Manifestations—Case Report

**DOI:** 10.3390/diseases10030062

**Published:** 2022-09-09

**Authors:** Céu Costa, Hugo Santiago, Sofia Pereira, Ana Rita Castro, Sandra Clara Soares

**Affiliations:** 1Escola Superior da Saúde, Universidade Fernando Pessoa, 4200-253 Porto, Portugal; 2Instituto de Investigação, Inovação e Desenvolvimento Fernando Pessoa, FP-I3ID (FP-BHS), 4249-004 Porto, Portugal; 3Grupo de Patologia Experimental e Terapêutica, Centro de Investigação, Instituto Português de Oncologia do Porto, 4200-072 Porto, Portugal; 4Faculdade de Ciências da Saúde, Universidade Fernando Pessoa, 4200-150 Porto, Portugal

**Keywords:** autoimmune diseases, multiple sclerosis, craniofacial manifestations, oral manifestations, oral health care, TMJ disorders, oral treatment, multidisciplinary approach

## Abstract

Background: Multiple sclerosis is an autoimmune disease of the central nervous system with neurological and motor symptoms that affect the orofacial region. The aim of this work is to present a patient that lacks the three classic orofacial manifestations but has other less common clinical alterations. Case presentation: A 49-year-old female patient diagnosed with long-term relapsing–remitting multiple sclerosis visited the dentist complaining of mild but persistent orofacial pain including the temporomandibular joint and pain not specific to any tooth. She presented mucosal irritation, xerostomia, halitosis, and localized gingivitis. There was excessive wear of the upper and lower incisal edges and the occlusal faces of the upper canines and loss of six teeth due to caries. After a clinical oral examination, the diagnosis was temporomandibular joint disorder, gingivitis, dental hypersensitivity, bruxism, hyposalivation, xerostomia, and halitosis. Conclusions: Patients with multiple sclerosis present classic orofacial manifestations. Although these were not observed in this patient, she had others, such as gingivitis, tooth hypersensitivity, and bruxism. In addition, despite few studies associating a higher prevalence of caries with these patients, the number of carious and missing teeth in this patient highlight the evidence that multiple sclerosis has had a significant impact on the patient’s dental status over the years.

## 1. Introduction

Multiple sclerosis (MS) is an immune-mediated disease that involves recurrent inflammation of the central nervous system resulting in damage to both the myelin sheath surrounding axons and the axons themselves. Histologic examination reveals foci of severe demyelination, decreased axonal and oligodendrocyte numbers, and glial scarring [1].

The aetiology of MS remains unknown. However, like most autoimmune diseases, it is considered multifactorial, involving environmental factors such as viral infections, smoking, vitamin deficiencies, especially vitamin D, and genetic predisposition. It affects women more than men at a ratio of 2.5 to 1 and is usually detected between the ages of 20 and 40 [2].

The clinical manifestations of MS vary from patient to patient leading to neurological and motor symptoms, such as fatigue, vision changes, sensory changes in the arms, legs, or face (prickling/burning sensation), urinary and intestinal dysfunction, motor problems (weakness, balance, and spasms), depression and anxiety, cognitive problems, speech difficulties, dysphagia, and pain. The latter can be acute pain with relapses or chronic pain with spasticity [3]. It is a complex disease that causes progressive disability and negatively affects the patient’s quality of life, especially their walking ability [4].

Disease activity and progression characterize the three clinical phenotypes of MS: in relapsing–remitting MS (RRMS), the most prevalent, the patient has episodes of disease exacerbation followed by periods of remission. These episodes are usually reversible but can cause permanent neurological deficits. In primary progressive MS (PPMS), the disease has a steady and permanent onset of the symptoms without remission periods. Secondary progressive MS (SPMS) develops from RRMS and is characterized by a stable disability worsening without recovery. In addition, the non-progressive or benign form is very rare and people live with a long-term absence of any disability [5,6].

Several drugs have been approved for MS therapy; these are mostly disease-modifying drugs such as interferon-β glatiramer acetate and those that act indirectly by controlling neuro-inflammation and MS symptoms. They are, thus, conventional therapies that are unable to stop the neurodegeneration [7,8].

Clinical manifestations of MS also arise in the orofacial region, in particular in the form of trigeminal neuralgia, usually one of the first manifestations of the disease, trigeminal sensory neuropathy, and facial paralysis [9].

Pain in the orofacial region, resembling pain of dental origin, is characterized by paroxysmal, throbbing pain similar to an electric shock. Paraesthesia, hemifacial spasms, and Charcot’s triad are also observed. The prevalence of periodontal disease is high in these patients; however, studies are contradictory regarding the prevalence of caries [10]. A previous study shows a relationship between chronic periodontitis and MS in female patients [11].

There is a high prevalence of temporomandibular joint (TMJ) disorders in MS, such as pain and difficulty opening the mouth and TMJ sounds, which can be attributed to myofascial and neck pain. Some studies also report teeth grinding, dental hypersensitivity, and xerostomia [12,13].

The oral health of patients with MS is also directly associated with the drugs classically used in the treatment of the disease, such as corticosteroids, muscle relaxants, anticonvulsants, antidepressants, anticholinergics, and immunosuppressants. Side effects that may affect the oral mucosa include xerostomia, gingival hyperplasia, mucositis, stomatitis, dysgeusia, candidiasis, and angular cheilitis [14]. Furthermore, oral hygiene in patients with MS can be limited due to neurological deficits, such as motor deficits, cognitive dysfunctions, visual disorders, and pain. Thus, the patients can report a good oral hygiene and not have a real cognitive perception [15].

Alcohol and smoking habits are known to interfere with the oral health of patients with MS, the same as in the general population [16,17,18].

This case report aims to present a patient diagnosed with long-term MS, with none of the three classic orofacial manifestations. Instead, the patient presented TMJ disorders, tooth hypersensitivity, and many teeth lost due to caries, gingivitis, xerostomia, bruxism, and halitosis. The last two are rarely associated with MS.

## 2. Case Report

A 49-year-old female patient diagnosed 17 years ago with relapsing–remitting MS (RRMS) visited the dentist complaining of orofacial pain and pain not specific to any tooth. During anamnesis and regarding the clinical history, the patient reported having no pathology other than MS and no relevant past medical history. The outbreaks occur only once a year, last about a week, and are controlled with corticosteroids.

At the time of examination, she was medicated for MS with Fampyra (10 mg, twice a day/morning and evening); Fingolimod FTY720 (0.5 mg, once a day/morning); Pregabalin (75 mg, once a day/morning). She was also taking other types of medication not directly related to the pathology: Omeprazole (20 mg, once a day/morning) and Eutirox (125 mcg, once a day/morning). The patient was also asked to complete a questionnaire about her oral hygiene habits.

A clinical oral examination revealed oral mucosal irritation, xerostomia, halitosis, and localized gingivitis (Figure 1).

One caries (tooth 45) and eight restorations were observed, two of which had infiltrations (teeth 34 and 44). She had lost six teeth due to caries, and the remaining had a normal morphology and size without mobility. There was excessive wear of the incisal edges of the upper and lower incisors and that of the occlusal faces of the upper canines (Figure 2).

The patient complained of mild but persistent head and neck pain including TMJ (Figure 3) in addition to pain not specifically related to any tooth. In the anamnesis and questionnaire, the patient mentioned gingival bleeding during brushing (twice a day) and frequent teeth grinding.

The extraoral clinical examination revealed facial symmetry, no palpable submandibular adenopathy, and the TMJ was found to be free of any deviation, clicking, or noise, despite the reported pain. The gingival margin was symmetrical with the midline centred.

The diagnosis was TMJ disorder, the most common cause of orofacial pain and gingivitis, as well as dental hypersensitivity. She was also diagnosed with bruxism and hyposalivation, which caused the xerostomia. Unstimulated salivary flow was assessed by collecting the patient’s saliva for a period of 15 min, at least two hours after eating, and measuring it in a syringe [19]. The flow was expressed in ml/min. The unstimulated salivary flow measurement was 0.03 mL/min, well below the normal range of 0.1 mL/min.

Bacterial plaque was removed with ozone therapy in the gum inflammation zone; the cavity of tooth 45 was restored and the patient was advised to use an oral irrigator to prevent further caries; fluoride paste was prescribed as well as mouthwash and brushes to reinforce oral hygiene. An occlusion adjustment was considered with implant placement of tooth 14 also for relief of orofacial pain and control of bruxism.

In the case of xerostomia and hyposalivation, oral hygiene instructions play a fundamental role, as well as adequate hydration (the recommendation is two litres of water per day and the ingestion of non-sugary liquids). Avoiding smoking and alcohol consumption are important, but the patient reported that these were not her habits.

After endodontic treatment, the patient asked about the possibility of placing more implants due to the high number of missing teeth (a total of 10) and veneers (Figure 4a,b). The patient is awaiting two results: a bone evaluation as to the feasibility of using implants and a new gingival evaluation in the case of veneers. In patients with MS, dental implants are a good alternative to removable prosthesis, avoiding slipping or displacement that can occur due to xerostomia.

The patient has shown interest in having aesthetic harmonization of the smile, depending on the cost of the planned treatment.

## 3. Discussion

MS is a chronic autoimmune disease of the central nervous system; the brain (especially the white matter) and spinal cord are affected because of the demyelinating inflammatory process that leads to the destruction of the myelin sheath of axons [20].

Areas of myelin loss, zones of scarring changes (sclerotic), are called plaques. These vary from 1 to 2 mm to 2 cm in diameter, and the conduction of the nerve impulse becomes ineffective, leading to neurological signs and symptoms [21].

Most patients with RRMS recover from the relapses, but the neurological deterioration gradually affects several systems, and the symptoms can vary from limb numbness/weakness to electric-shock sensations, vision problems, fatigue, and tremors to an inability to walk [3].

Patients with MS present three common orofacial manifestations: trigeminal neuralgia, trigeminal paraesthesia, and facial palsy. None of these conditions were observed in this patient, but others, such as TMJ disorders, tooth hypersensitivity, gingivitis, hyposalivation, and (less common in MS) caries and (even more rarely) bruxism and halitosis, were identified.

In this clinical case, TMJ disorders are evident, agreeing with the literature which indicates that people with MS have a greater susceptibility [12]. TMJ disorders are one of the most common causes of orofacial pain in patients with MS, and they have a multidimensional effect, with pain symptoms in the pre- and post-auricular areas, mandibular angle, and ramus and temporal region. The pain usually radiates to the temporal, occipital, or cervical regions and in the malar region of the face; it can be unilateral or bilateral and of variable severity, usually aggravated by opening the mouth, yawning, or chewing [22]. Education, psychological support, and self-management strategies are recommended as part of a multidisciplinary approach to TMJ management and should be performed early to prevent worsening of symptoms [23].

The patient has dental hypersensitivity, which is in line with the literature that states that people with MS are approximately three times more susceptible to this symptom [24].

Currently, only one study points to the higher prevalence of bruxism in people with MS [25], and in the case of this patient, it could certainly be one of the causes of her ongoing head and neck pain. The sensory and motor disorders of MS can affect a person in several ways, and the risk of onset and progression of bruxism increases when occlusion is impaired [26], as was observed. Previous studies linked jaw clenching/bruxism and temporal bone movement in patients with MS. The displacement of cranial bones can cause fluid pressure changes in the ventricles and damage brain tissue around them [27]. As a result of this mechanical stress, osteoclast-mediated bone resorption can be accelerated [28], creating conditions for caries to install. Thus, by itself, bruxism may also be a contributor to the increased prevalence of caries, despite the existence of few studies associating a higher prevalence of caries in patients with MS [29]. In this clinical case, the number of teeth missing due to dental caries highlights the evidence that MS has had a significant impact on the patient’s dental status over the years. Imaging examinations would be necessary to confirm this association.

One of the first symptoms of gingivitis is bleeding gums, followed by erythema. The degree of gingival bleeding and the ease with which it is provoked depend on the severity, duration of the inflammatory process, and the intensity of the triggering factors [30]. Gingival and periodontal diseases are more prevalent in people with MS. This patient reported the absence of risk factors, such as alcohol or tobacco consumption, and stated having a good oral hygiene with dental appointments every 6 months. However, in MS, the mechanism associated with the development of gingivitis and periodontitis can be related not only to the patient’s ability to perform, or not, a proper oral hygiene but also with the false perception of performing this hygiene properly [11].

Xerostomia is associated with a lack of saliva and may underlie several medical conditions and/or the use of certain drugs. In this clinical case, xerostomia may be caused by the disease itself, as studies indicate [24], or by the association of substances used to manage the disease. The anticonvulsant Pregabalin can cause xerostomia [31]. Another drug used by the patient is Fampyra, whose active compound of extended-release Fampridine is currently the only drug approved to treat walking impairment in adult patients with MS. It acts by blocking potassium channels that can also be found in the salivary glands. Thus, it can cause a decrease in the salivary flow [32]. The drug Omeprazol, a gastric protector, acts by irreversible inhibition of the hydrogen/potassium adenosine triphosphatase enzyme and can also cause a decrease of the salivary production [33]. In summary, the combination of drugs used by the patient is likely to be the cause of hyposalivation and xerostomia. Another consequence of hyposalivation may be the halitosis presented by the patient, although dysphagia and dysgeusia, two other conditions associated with xerostomia, were not part of the patient’s clinical profile.

The presence of orofacial classic manifestations is significantly correlated with patients with more than 7 years of diagnosed MS disease [9,34]. In the present case, the patient was diagnosed 17 years ago, surprisingly with no manifestations of this type at this point.

One possible explanation can be the number of outbreaks per year that occurs in this patient. According to Zhang, the average is about three relapses/year [34], but in the present case, only one was reported per year with duration of a week. Probably, this event caused a delay in the appearance of classic manifestations, allowing other disorders to occur. The absence of alcohol and smoking habits can positively impact her oral cavity health.

The medical dentist plays an important part in the diagnosis and management of the oral conditions of patients with MS as part of a multidisciplinary team. Seeing that MS is a dynamic disease, producing multifocal neurological deficits and disability, a wide range of healthcare specialists are needed to assist in MS care throughout the patient’s life. The neurologist is typically the primary care provider, but the need for other specialists, such as dentists, psychologists, ophthalmologists, speech and language pathologists, and social workers, is common [35,36].

## 4. Conclusions

Several types of oral lesions and orofacial manifestations are observed in patients with MS due to the disease itself or the medications used to manage it. In this case, the typical orofacial manifestations—trigeminal neuralgia, trigeminal paraesthesia, and facial palsy—are absent. However, other manifestations can be identified: TMJ disorders, tooth hypersensitivity, and gingivitis. Bruxism, although rare, was also diagnosed, as was halitosis despite the patient claiming to have a good oral hygiene. The number of carious missing teeth was high, an association that is not very common in this disease. This can be a direct consequence of hyposalivation and highlights the evidence that this disease has had a significant impact on the patient’s dental status over the years. The number of outbreaks per year and absence of risk factors, such as alcohol and tobacco, can explain why manifestations other than the classic ones occur and can help in future studies of this specific phenotype of MS and oral health manifestations.

This work intends to alert to the importance of autoimmune inflammatory diseases in the world of dentistry. In the case of MS, the impact on oral health is visible, even with non-classic manifestations, so it is recommended that the dentist establish an appropriate treatment plan. Treatments, thus, must be well organized; simultaneously, the multidisciplinary team of neurologists and rheumatologists, among others, must be on the alert to avoid possible complications of the disease itself or of the drugs prescribed. It is also important to plan interventions that, although they may improve the patient’s oral health in the short term, may present later complications.

## Figures and Tables

**Figure 1 diseases-10-00062-f001:**
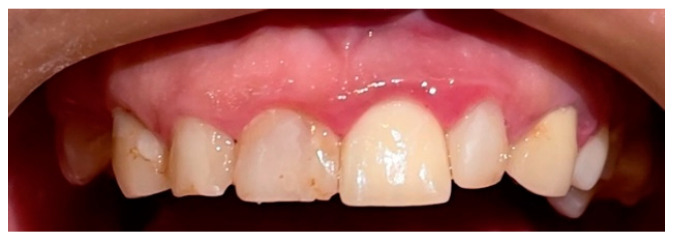
Superior arch localized gingivitis: redness, oedema, and modification of the normal teeth contour.

**Figure 2 diseases-10-00062-f002:**
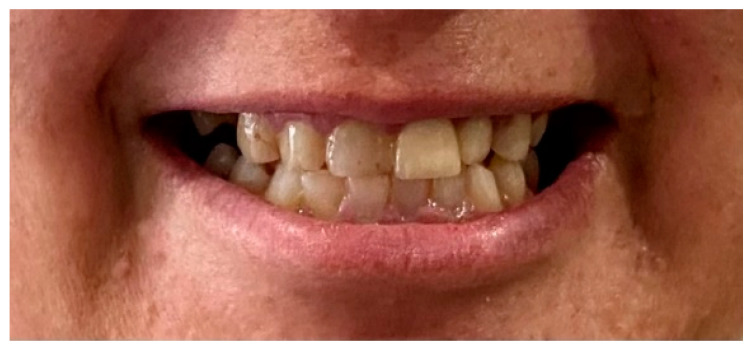
Upper and lower incisors and canines wear.

**Figure 3 diseases-10-00062-f003:**
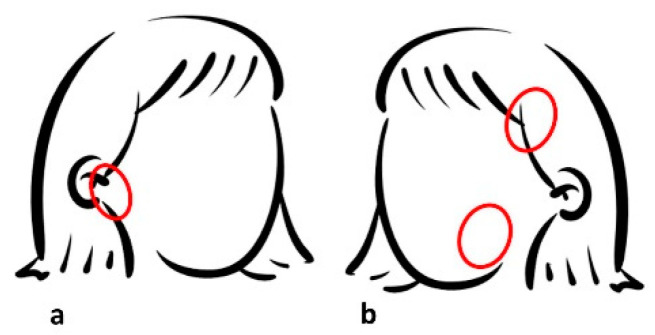
Pain locations across the head and neck region: (**a**) temporomandibular joint; (**b**) craniofacial area.

**Figure 4 diseases-10-00062-f004:**
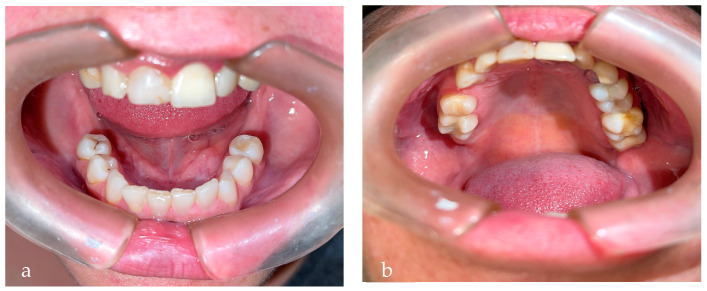
(**a**) Inferior arch missing teeth: 36, 37, 38, 46, 47, and 48; (**b**) superior arch missing teeth: 14, 17, 18, and 28.

## Data Availability

Data is contained within the article.

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
