# Peer review of "Oral Health Status and Multiple Sclerosis: Classic and Non-Classic Manifestations—Case Report"

_diseases, 2022, doi:10.3390/diseases10030062_

Round 1

Reviewer 1 Report

Dear authors it’s an interesting article, 

but please add an abstract.

also there are number of sentences like: “The medical dentist is an important part in the diagnosis and” he probably mean he has an important role in or plays an important part

“In summary the combination of drugs used by the patient is certainly the cause of hyposalivation and xerostomia.” Please change certainly to likely because you do not present any direct proof.

You state that: “MS patients present three common orofacial manifestations: trigeminal neuralgia, trigeminal paresthesia and facial palsy.” Yet a few lines later you say that: “TMJ disorders are the most common cause of orofacial pain in patients with MS”. That doesn’t make any sense in line of your first  statement because the word neuralgia in “trigeminal neuralgia” means that patients have pain in the distribution of that nerve.

Author Response

Manuscript ID diseases-1865602

“Oral health status and Multiple Sclerosis:  classic and non classic manifestations”

("Pending Major Revisions")

Reviewers 1 and 3 mention “ Moderate English changes required” and reviewer 4 “English language and style are fine/minor spell check required” so the paper was revised by one of the members of the University - The English Teacher, a native speaker.

Table 1- Reviewer 1 comments and response

add an Abstract

The Abstract was inserted by mistake in a different part of the MDPI template, but it is found in the end of the document -line 291.

It`s now in the beginning of the article.

Unclear sentence:

“The medical dentist is an important part in the diagnosis…….”

We agree with the comment and changed the sentence to:

“The medical dentist plays an important part in the diagnosis………”

Unclear sentence:

“In summary the combination of drugs used by the patient is certainly the cause of hyposalivation and xerostomia.” Please change certainly to likely because you do not present any direct proof.

We agree with the comment and changed the sentence to:

“In summary the combination of drugs used by the patient is likely to be the cause of hyposalivation and xerostomia.”

You state that: “MS patients present three common orofacial manifestations: trigeminal neuralgia, trigeminal paresthesia and facial palsy.” Yet a few lines later you say that: “TMJ disorders are the most common cause of orofacial pain in patients with MS”. Line 126

That doesn’t make any sense in line of your first  statement because the word neuralgia in “trigeminal neuralgia” means that patients have pain in the distribution of that nerve.

We agree with the reviewer and changed the sentence to:

“TMJ disorders are one of the most common cause of orofacial pain in patients with MS……..”

We are thankful for your sugestions/comments.

Reviewer 2 Report

The authors reported an interesting case of multiple sclerosis. I don’t have major criticism about the finding, but some comments about the format should be addressed.

1. Please use the template of MDPI. You can find it from the website or can get help from assistant editor.

2. The first letter of each word in the title should be capitalized.

3. Names: ‘and’ should be placed between ‘Ana Rita Castro’ and ‘Sandra Clara Soares’

4. I understand five authors contribute equal. Who is the corresponding author(s)?

5. Page 5, Ethics Approval and Informed Consent. Please provide the Informed Consent file.

6. Page 5, a lot of info is missing such as Acknowledgment, Data Availability Statement, Funding, and so on.

7. In the author contribution section, please specify the contribution of each author based on the requirement of MDPI.

Author Contributions: For research articles with several authors, a short paragraph specifying their individual contributions must be provided. The following statements should be used “Conceptualization, X.X. and Y.Y.; methodology, X.X.; software, X.X.; validation, X.X., Y.Y. and Z.Z.; formal analysis, X.X.; investigation, X.X.; resources, X.X.; data curation, X.X.; writing—original draft preparation, X.X.; writing—review and editing, X.X.; visualization, X.X.; supervision, X.X.; project administration, X.X.; funding acquisition, Y.Y. All authors have read and agreed to the published version of the manuscript.” Please turn to the CRediT taxonomy for the term explanation. Authorship must be limited to those who have contributed substantially to the work reported.

Author Response

Manuscript ID diseases-1865602

“Oral health status and Multiple Sclerosis:  classic and non classic manifestations”

("Pending Major Revisions")

Reviewers 1 and 3 mention “ Moderate English changes required” and reviewer 4 “English language and style are fine/minor spell check required” so the paper was revised by one of the members of the University - The English Teacher, a native speaker.

Table 2- Reviewer 2 comments and response

Does the introduction provide sufficient background and include all relevant references?

The reviewer says that this point can be improved.

Explanation about MS and references were add in the introduction to provide the needed background.

Are the methods adequately described?

The reviewer says that this point can be improved.

We agree with the reviewer and describe further the methodology used in the Case Report

Are the conclusions supported by the results?

The reviewer says that this point can be improved.

The authors altered the conclusion to better reflect the clinical results

Please use the template of MDPI. You

can find it from the website or can get help from assistant editor.

The authors agree and correct the editing text using the MDPI template accordingly.

The first letter of each word in the title should be capitalized.

The authors agree and correct

Names: ‘and’ should be placed between ‘Ana Rita Castro’ and ‘Sandra Clara Soares’

The authors agree and correct

I understand five authors contribute equal. Who is the corresponding author(s)?

The authors agree. It is already in the paper.

Céu Costa is the corresponding author

Ethics Approval and Informed Consent. Please provide the Informed Consent file.

The documents the reviewer mention were sent to MDPI-Diseases assistant editor.

A lot of info is missing such as Acknowledgment, Data Availability Statement, Funding, and so on.

The missing information is now described in the paper.

In the author contribution section, please specify the contribution of each author based on the requirement of MDPI.

This missing information is now available in the paper.

We are thankful for your comments/sugestions.

Reviewer 3 Report

The authors present an interesting and well-written clinical case. The manuscript has minimal points of improvement:

-First of all, the title must be revised, making it clear that it is a clinical case and a review of the literature.

-Secondly, the authors must add more keywords in the new version of the manuscript.

- The figure legends should be improved, the authors should give more information.

-The authors carry out an adequate discussion, but they should add a comparative table of the main authors in the area.

-The authors must adequately improve the use of English grammar.

Author Response

Manuscript ID diseases-1865602

“Oral health status and Multiple Sclerosis:  classic and non classic manifestations”

("Pending Major Revisions")

Reviewers 1 and 3 mention “ Moderate English changes required” and reviewer 4 “English language and style are fine/minor spell check required” so the paper was revised by one of the members of the University - The English Teacher, a native speaker.

Table 3- Reviewer 3 comments and response

Are all the cited references relevant to the research?

The reviewer says that this point can be improved.

The authors add more relevant references to the cited ones, in the introduction.

Are the results clearly presented?

The reviewer says that this point can be improved.

The authors agree and add more information to the image legends making the results clearer.

The title must be revised, making it clear that it is a clinical case and a review of the literature.

The tittle was changed to:

Oral Health Status and Multiple Sclerosis: Classic and Non-Classic Manifestations – Case Report

The authors must add more keywords in the new version of the manuscript.

The authors add the following keywords:

Autoimmune diseases; craniofacial manifestations and Oral manifestations

The figure legends should be improved, the authors should give more information.

The authors agree and the figure legends were changed including more detailed information

The authors carry out an adequate discussion, but they should add a comparative table of the main authors in the area

The authors understand the comment but found it clearer to present this comparison in the Discussion because it is a case report and not a Review article.

We are thankful for your sugestions/comments. 

Reviewer 4 Report

The manuscript reported Awareness about Oral health status and Multiple Sclerosis, provides a good explanation of the classical and non-classical clinical manifestations of Oral health status and Multiple Sclerosis (MS) and alert to the importance of autoimmune inflammatory diseases.

In general, this manuscript is well-organized and the authors' claims can be supported well by the investigation results and manifestations.The manuscript can be published in  with the following concerns are addressed:

1.     In the introduction part, the explanation of Multiple Sclerosis (MS) is not systematic and comprehensive, and still needs to be supplemented to this knowledge.

2.     This manuscript is a clinical study and explanation of oral health and non-classical clinical cases of multiple sclerosis (MS). The research and discussion of a single case should pay more attention to the case itself, such as the oral problems and living habits of the case, which have a great impact on the oral problems. To explore more common features of non-classical clinical manifestations. This is of great significance for the future study of such oral problems.

3.     In Figure 4, the annotation format does not match the image exactly.

Author Response

Manuscript ID diseases-1865602

“Oral health status and Multiple Sclerosis:  classic and non classic manifestations”

("Pending Major Revisions")

Reviewers 1 and 3 mention “ Moderate English changes required” and reviewer 4 “English language and style are fine/minor spell check required” so the paper was revised by one of the members of the University - The English Teacher, a native speaker.

Table 4 - Reviewer 4 comments and response

Does the introduction provide sufficient background and include all relevant references?

The reviewer says that this point can be improved.

The authors add more relevant references to the cited ones, in the introduction

In the introduction part, the explanation of Multiple Sclerosis (MS) is not systematic and comprehensive, and still needs to be supplemented to this knowledge.

Explanation about MS and references were add in the introduction

The research and discussion of a single case should pay more attention to the case itself, such as the oral problems and living habits of the case, which have a great impact on the oral problems. To explore more common features of non-classical clinical manifestations. This is of great significance for the future study of such oral problems.

The introduction and discussion were improved paying more attention to the habits of the patient and their impact in the oral problems.

In the discussion and conclusion, the authors pointed out the non-classical manifestations and their significance in future studies.

In Figure 4, the annotation format does not match the image exactly.

The authors agree, it was a formatting mistake, and it is now corrected.

We are thankful for your comments/sugestions. 

Round 2

Reviewer 2 Report

The authors reponsed and revised my comments. I would like to recommend to publish it.